# A Hybrid Epithelial to Mesenchymal Transition in Ex Vivo Cutaneous Squamous Cell Carcinoma Tissues

**DOI:** 10.3390/ijms23169183

**Published:** 2022-08-16

**Authors:** Christopher S. Pulford, Chandana K. Uppalapati, McKale R. Montgomery, Richard L. Averitte, Elizabeth E. Hull, Kathryn J. Leyva

**Affiliations:** 1Arizona College of Osteopathic Medicine, Midwestern University, 19555 N. 59th Avenue, Glendale, AZ 85308, USA; 2Biomedical Sciences Program, College of Graduate Studies, Midwestern University, 19555 N. 59th Avenue, Glendale, AZ 85308, USA; 3Department of Microbiology & Immunology, College of Graduate Studies, Midwestern University, 19555 N. 59th Avenue, Glendale, AZ 85308, USA; 4Nutritional Sciences, 416 Nancy Randolph Davis, Stillwater, OK 74078, USA; 5Affiliated Dermatology & Affiliated Laboratories, 20401 N. 73rd Street #230, Scottsdale, AZ 85255, USA

**Keywords:** cutaneous squamous cell carcinoma, E-cadherin, hybrid EMT, TGF-β, vimentin

## Abstract

While most cases of cutaneous squamous cell carcinoma (cSCC) are benign, invasive cSCC is associated with higher mortality and is often more difficult to treat. As such, understanding the factors that influence the progression of cSCC are important. Aggressive cancers metastasize through a series of evolutionary changes, collectively called the epithelial-to-mesenchymal transition (EMT). During EMT, epithelial cells transition to a highly mobile mesenchymal cell type with metastatic capacities. While changes in expression of TGF-β, ZEB1, SNAI1, MMPs, vimentin, and E-cadherin are hallmarks of an EMT process occurring within cancer cells, including cSCC cells, EMT within tissues is not an “all or none” process. Using patient-derived cSCC and adjacent normal tissues, we show that cells within individual cSCC tumors are undergoing a hybrid EMT process, where there is variation in expression of EMT markers by cells within a tumor mass that may be facilitating invasion. Interestingly, cells along the outer edges of a tumor mass exhibit a more mesenchymal phenotype, with reduced E-cadherin, β-catenin, and cytokeratin expression and increased vimentin expression. Conversely, cells in the center of a tumor mass retain a higher expression of the epithelial markers E-cadherin and cytokeratin and little to no expression of vimentin, a mesenchymal marker. We also detected inverse expression changes in the miR-200 family and the EMT-associated transcription factors ZEB1 and SNAI1, suggesting that cSCC EMT dynamics are regulated in a miRNA-dependent manner. These novel findings in cSCC tumors provide evidence of phenotypic plasticity of the EMT process occurring within patient tissues, and extend the characterization of a hybrid EMT program occurring within a tumor mass. This hybrid EMT program may be promoting both survival and invasiveness of the tumors. A better understanding of this hybrid EMT process may influence therapeutic strategies in more invasive disease.

## 1. Introduction

Non-melanoma skin cancers, such as basal cell carcinoma (BCC) and cutaneous squamous cell carcinoma (cSCC), are the most common forms of cancer in the US. An estimated 3.5 million cSCCs and BCCs are diagnosed each year, and the incidence of these types of cancer continues to increase [1,2]. Cutaneous SCCs are less common but are more invasive and more likely to metastasize than BCCs. Fortunately, >95% of cSCCs are treated successfully by tumor excision, but prognosis is poor when cSCC metastasizes [3]. However, because of this, cSCC research is scarce and therapies for the ~5% of advanced cases that metastasize are limited, resulting in a 5-year survival rate as low as 25–35% [3,4,5]. Thus, studies that will identify molecular biomarkers for disease assessment, surveillance, and therapy of cSCC are warranted and needed.

Cancer metastasis is multi-factorial, including a tumor cell epithelial-to-mesenchymal transition (EMT). During EMT, epithelial cells undergo a de-differentiation process whereby they transition to a highly mobile and invasive mesenchymal cell type with metastatic capacities [6,7,8,9,10]. Within cancerous tissues, EMT has been shown to be induced by growth factors and cytokines, such as TGF-β. TGF-β is well known to be a positive regulator of tumor progression and metastasis (e.g., [8,9,11,12]). Additionally, these cells within the tumor undergoing EMT exhibit a newly described phenotypic heterogeneity not found in nearby non-cancerous tissue [10].

The EMT process involves the disruption of cell–cell adhesion and cellular polarity, remodeling of the cytoskeleton, and changes in cell–extracellular matrix adhesion. It has been noted that a complete EMT process rarely occurs in human cancer tissues; in fact, EMT is often activated reversibly, reverting back to an epithelial state during cancer progression or establishment of metastatic sites [10]. In some of these cancer tissues, cells at the leading edge (invasive front) of the tumor exhibit signs of EMT activation, including the reduced expression of E-cadherin, while cells that follow behind usually display many epithelial traits and maintain extensive cell–cell adhesions [8,13]. In vivo, carcinomas concomitantly express a hybrid of both epithelial and mesenchymal characteristics, rarely losing all their epithelial traits [8,9,10]. More recently, it has been exhibited that the extent of EMT does not correlate with metastasis [8,14].

We hypothesize that during progression of cSCC, a hybrid EMT process is occurring within cSCC tumors. Our data add to the growing body of literature that the EMT process within carcinoma tissues may not follow a typical Type 3 EMT process associated with cancer progression initially described by Kalluri and Weinberg [15], where cells undergoing EMT decrease E-cadherin expression with a corresponding increase in vimentin expression. As recently pointed out by Fernandez-Figueras and Puig [16], information about the specific EMT process occurring in primary cSCC is sparse. Our data support our hypothesis that a hybrid EMT process is occurring within cSCC tumors. As phenotypic diversity of cells within a tumor often creates challenges to successful treatment [10], more knowledge is needed regarding the complexity and heterogeneity of EMT occurring in carcinomas and its control of cancer progression and metastasis.

## 2. Results

### 2.1. Patient-Derived cSCC Tissues Exhibit Increased mRNA Expression of TGF-β and E-Cadherin

Transforming growth factor-β (TGF-β) is a well-known inducer of EMT in many biological systems. We assessed mRNA expression of TGF-β and TGF-βR2 in our patient-derived cSCC tissues. Compared to adjacent normal tissues (ANTs), cSCC tissues exhibited a 27-fold increase in TGF-β mRNA (*p* < 0.05, N = 5) and a 100-fold increase in TGF-βR2 mRNA (*p* < 0.001, N = 5) (Figure 1A,B). We also measured levels of the epithelial marker E-cadherin (CDH1), which is downstream of TGF-β in the EMT process. Interestingly, we found E-cadherin mRNA expression was 3.5-fold higher in our cSCC tissues compared to ANT (*p* < 0.005, N = 5) (Figure 1C).

### 2.2. EMT-Promoting Protein Expression and Total MMP Activity Are Higher in cSCC Tissues

We performed a Proteome Profiler Human XL Oncology Array to assess expression levels of 84 oncogenic proteins within cSCC tissues compared to ANT. We observed that patient-derived cSCC tissues exhibited significantly higher levels of 13 different oncogenic proteins (Appendix A); 12 of these 13 proteins, including matrix metalloproteinase 2 (MMP2), are known to promote EMT in various cancers. To determine if the observed increase in MMP2 expression is correlated to activity, we assessed MMP activity using a FRET assay. We observed that total activity of MMPs within patient-derived cSCC tissue samples was 2.5-fold higher, measured at 2 h (*p* < 0.01, N = 13), and 2-fold higher, measured at 4.5 h (*p* < 0.001, N = 13), compared to ANT (Figure 2).

### 2.3. Expression of Epithelial and Mesenchymal EMT Markers Is Increased in cSCC Tissues

We next examined expression level of EMT markers based on cSCC invasiveness; tissues were categorized as either invasive or non-invasive. In our invasive cSCC tissue samples, we observed a statistically significant increase in expression of four known EMT-associated proteins: TGF-β, E-cadherin, phosphorylated E-cadherin, and vimentin. Compared to ANT, non-invasive cSCC tissue expression of TGF-β was 3.2-fold higher (*p* < 0.05, N = 13), E- cadherin was 3-fold higher (*p* < 0.05, N = 8), phosphorylated E-cadherin was 5.4-fold higher (*p* < 0.001, N = 8), and vimentin was 1.4-fold higher (*p* < 0.05, N = 17) (Figure 3A,B). Compared to ANT, invasive cSCC tissue expression of TGF-β was 14-fold higher (*p* < 0.001, N = 11), E-cadherin was 7.5-fold higher (*p* < 0.05, N = 8), phosphorylated E-cadherin was 17-fold higher (*p* < 0.001, N = 8), and vimentin was 7-fold higher (*p* < 0.005, N = 10) (Figure 3C,D).

### 2.4. There Is Differential E-Cadherin Expression among Cells within cSCC Tumors

To address this paradox of increased E-cadherin contrary to what is observed in classical EMT, we examined expression of E-cadherin (Figure 4A) and phosphorylated E-cadherin (Figure 4B) using immunofluorescent assays performed on intact cSCC tissue and ANT. We confirmed that overall protein expression of both E-cadherin (*p* < 0.001, N = 64) and phosphorylated E-cadherin (*p* < 0.0001, N = 106) was significantly higher in cSCC tissues compared to ANT (Figure 4C). Interestingly, we noted that this increased expression was not uniform within a tumor mass. Examination of cells within distinct regions of a tumor mass revealed that expression of E-cadherin was significantly higher in well-differentiated cells located in the center of an individual tumor mass compared to the poorly differentiated cells located along the outer edges (outer 1–3 layers of cells) of an individual tumor mass (Figure 4D).

### 2.5. Coexpression of E-Cadherin and β-Catenin Is Lost in cSCC Tumor Cells Exhibiting EMT

As β-catenin is an important component in E-cadherin adherens junctions, we next wanted to determine if expression of β-catenin mirrored the pattern we observed with E-cadherin. Dual staining of cSCC tissue sections with antibodies against E-cadherin (green; Figure 5, left panels) and β-catenin (red; Figure 5, middle panels) showed that these two proteins are membranous and appear to be coexpressed in well-differentiated cells (yellow; Figure 5, right panels), which is indicative of a more epithelial phenotype. We observed a significant decrease in expression, measured as mean ± SEM fluorescent intensity (in arbitrary units), of both E-cadherin (1.3 ± 0.12 × 10^6^, *p* = 0.048, N = 10) and β-catenin (1.4 ± 0.12 × 10^6^, *p* = 0.043, N = 10) in the poorly differentiated cells compared to the well-differentiated cells (2.3 ± 0.46 × 10^6^ and 2.5 ± 0.43 × 10^6^, N = 10, respectively), indicative of the poorly differentiated cells having a more mesenchymal phenotype.

### 2.6. The Expression Pattern of Vimentin within an Individual Tumor Is Reversed Compared to E-Cadherin

We performed immunohistochemistry on ANT and cSCC using a pan-cytokeratin antibody to identify the cells that are epithelial in nature. Both ANT (Figure 6A) and cSCC tissues (Figure 6B,C) stained robustly red, with fibrous tissue and immune cells remaining unstained for pan-cytokeratin. Not surprisingly, we did not observe any significant difference in pan-cytokeratin expression, measured as mean ± SEM optical density, between cSCC (0.35 ± 0.04, *p* = 0.102, N = 6) and ANT (0.26 ± 0.03, N = 5) as both tissue types have an abundance of epithelial tissue. However, at higher magnification (100×), it was strikingly evident that the poorly differentiated cells at the tumor edges stained visibly lighter than the well-differentiated cells in the center of the tumor (Figure 6C). We performed immunohistochemistry using the mesenchymal marker vimentin on these same ANT and cSCC tissue sections (Figure 6D–F) to determine if a similar pattern was observed. We observed a significant increase in vimentin expression, measured as mean ± SEM optical density, in cSCC tissues (0.261 ± 0.026, *p* = 0.015, N = 6) compared to ANT (0.045 ± 0.001, N = 5). Remarkably, there was more intense vimentin staining (2–3+) in the poorly differentiated cancer cells (poor) at the edges of tumors (Figure 6F) than in the well-differentiated cells (well; 0–1+).

### 2.7. Expression of ZEB1 and SNAI1 Is Higher in cSCC Tissues

Next, we stained the commercially obtained tissue sections to determine the expression and distribution of ZEB1 and SNAI1, two transcription factors associated with promoting EMT (Figure 7). We observed a significant increase in expression, measured as mean ± SEM fluorescent intensity (in arbitrary units), of both ZEB1 (4.0 ± 1.1 × 10^6^, *p* = 0.011, N = 6) and SNAI1 (6.2 ± 2.2 × 10^6^, *p* = 0.034, N = 6) in cSCC tissues when compared to ANT (0.3 ± 0.1 × 10^6^ and 1.0 ± 0.6 × 10^6^, N = 5, respectively). Higher magnification shows that expression of these transcription factors is localized to the nucleus (Panels D and H). We confirmed these results by immunohistochemistry, which showed an increased expression of both transcription factors in cSCC (3+) compared to ANT (0–1+) tissues (Appendix A).

### 2.8. Hybrid EMT-Associated miRNAs Are Differentially Expressed in cSCC Tissues

The complex network of transcription factors, miRNAs, epigenetic modulators, and environmental signals that regulate the EMT program also allows for partial transitions of a tumor mass into a hybrid EMT phenotype that exhibits both epithelial and mesenchymal properties. Thus, we examined whether miRNA regulation may be contributing to this EMT phenotype observed in our cSCC tissues. Indeed, we observed that miR-34 as well as miR-200b and miR-200c were significantly downregulated (*p* < 0.05, N = 6) in cSCC compared to ANT (Figure 8). The expression of the oncogenic miRNA miR-31, known to promote cSCC invasiveness, was significantly increased (*p* < 0.001, N = 6) in our cSCC tissues as well (Figure 8).

## 3. Discussion

The epithelial to mesenchymal transition (EMT) is a plastic process by which epithelial cells lose their adhesiveness and polarity, gaining migratory and/or invasive properties characteristic of mesenchymal cells. Of the EMT types defined by Kalluri and Weinberg [15], many cancer tissues exhibit Type 3 EMT, where genetic and epigenetic changes in cancerous cells may affect expression of EMT proteins to facilitate invasion and metastasis in a different manner than the other two types of EMT (i.e., organogenesis and wound healing).

EMT may not be an “all or none” response. Historically, EMT has been regarded as a binary process, with cells existing in either an epithelial or a mesenchymal state [17]. During classic EMT, the epithelial marker E-cadherin is often reduced or lost while the mesenchymal marker vimentin is upregulated. However, this may not be occurring in all cells within a tumor, as our data support. As pointed out in recent reviews [17,18,19], the EMT process is dynamic, possibly including intermediate states that may be missed by assessment of a few markers at the beginning and end stages of EMT. Pastushenko et al. [19] highlight that in vivo data to support this idea are lacking. We provide insight into this dynamic process through using patient-derived squamous cell carcinoma tissues, rather than established cell lines, to characterize the type of EMT that is occurring within cutaneous squamous cell carcinoma. Our data revealed reduced expression of the epithelial markers E-cadherin and β-catenin, with a corresponding increase in the mesenchymal marker vimentin, in only leading edges of a tumor mass. These novel findings indicate that individual cells within an cSCC tumor mass exhibit phenotypic plasticity that may promote invasiveness and metastasis.

Collectively, our data support that a hybrid EMT phenotype is occurring within cSCC tissues. Twelve of the 13 differentially expressed proteins in our patient-derived cSCC tissues have been shown to promote EMT in other types of cancer. For example, tenascin C induces EMT in breast cancer cells [20], endoglin regulates EMT in renal cell carcinoma [21], carbonic anhydrase IX promotes EMT in prostate cancer cells [22], and osteopontin has been credited as a master regulator of EMT [23]. Additionally, the more classic EMT markers, such as TGF-β and TGF-βR2, were significantly upregulated as expected. TGF-β is known to be one of the main inducing signals of EMT in cancer [24,25,26,27,28]. Not only was TGF-β elevated within the tissues, but we also saw higher expression of its receptor, TGF-βR2, which is suggestive of an overall increase in TGF-β-mediated signaling within the cancer tissues [26]. Our data also confirm that two EMT-associated transcription factors, ZEB1 and SNAI1, are upregulated within cSCC tissues, especially in the poorly differentiated cells at the leading edges of the tumor mass, as shown in Figure 7 and Appendix A. It is well established that TGF-β-mediated signaling upregulates vimentin expression in epithelial cells undergoing EMT [24,28,29]. Furthermore, TGF-β-mediated signaling induces expression of MMPs, notably MMP2 and MMP9, potentially facilitating the migration and invasion of cancer cells (reviewed in [28,30]). We demonstrated that MMP activity was significantly higher in cSCC tissues. While we did not measure specific MMPs directly, the increased TGF-β-mediated signaling supports the idea that our increased MMP activity is associated with an EMT process within these tissues.

TGF-β-mediated signaling is also typically associated with decreased expression of E-cadherin (reviewed in [28]), which is in opposition to the pattern we observed. Surprisingly, we found that E-cadherin mRNA and protein expression were significantly increased in both non-invasive and invasive cSCC tissues relative to their respective ANT controls. While elevated E-cadherin levels have been reported in some cancer cell lines showing evidence of EMT [31,32] and reviewed in [33], these findings initially appeared inconsistent with an EMT process. As these Western blots were performed on bulk lysates from tumors and thus contain a heterogenous mix of cell types, we used immunofluorescent analysis of intact tissue sections to explain these interesting results. We observed distinct differences in cellular expression of E-cadherin within an individual tumor. In stable non-cancerous epithelial cells, E-cadherin is localized to adherens junctions within the cellular membrane, and is typically phosphorylated at serine residues 840, 846, and 847, promoting β-catenin binding and stabilizing the cadherin/catenin complex [34]. In cSCC tissue sections, only the poorly differentiated cells at the leading edges of the tumor express significantly lower levels of membranous E-cadherin and phosphorylated E-cadherin, and also exhibit a significant reduction in E-cadherin/β-catenin coexpression, compared to well-differentiated tumor cells, which would have been overlooked if only Western blot analysis of total protein expression was used. The increase in overall E-cadherin levels can be seen in the cross-section of these tissues, as the abundance of well-differentiated cells greatly outweighs the number of cells at the leading edge. Analysis of homogenized whole tissues, or in vitro cell line experimentation, misses this cell-dependent variation in EMT within these tumors; the use of intact, patient-derived tissues allowed us to detect this distinct patten of variation with individual tumor masses.

The hybrid EMT phenotype within our cSCC tissues has also been observed in several cancer models of collective cell migration [35,36,37] and reviewed in [17]. This hybrid phenotype is characterized with cells at the leading edge acquiring mesenchymal features while neighboring cells maintain intact cell–cell adhesions. Within our cSCC tissues, we have shown that the poorly differentiated cells at the edges of the tumors exhibit a mesenchymal morphology, expressing vimentin with a corresponding loss of pan-cytokeratin and membranous E-cadherin. Additionally, the poorly differentiated cells show a loss of membranous E-cadherin/β-catenin coexpression and increased cytoplasmic E-cadherin levels, which other studies have shown are consistent with tumor proliferation and invasiveness [38,39,40,41], reviewed in [30]. The well-differentiated cells within the tumor mass do not express vimentin, and abundant, stable membranous E-cadherin/β-catenin adhesions are maintained as evidenced by high levels of phosphorylated E-cadherin supporting this association. Our results clearly show the variation in expression of these EMT markers is not occurring at the single-cell level, but rather can be explained by the heterogeneity of the degree of cellular differentiation within distinct areas of the tumor mass. A limitation of this study, imposed by the tissues we obtained, is that we were unable to observe if hybrid EMT was occurring at the single-cell level. However, we do show that the heterogenous cancer cell population within a single tumor displays a spectrum of epithelial-to-mesenchymal markers that are consistent with a hybrid EMT state in cSCC.

EMT plasticity is critical for cancer metastasis as cells must be able to revert to their epithelial state to successfully colonize a secondary tissue. Unfortunately, the epithelial–mesenchymal plasticity observed in hybrid EMT phenotypes creates a “stemness window” that promotes the maintenance of metastasis-initiating cells that can worsen cancer prognosis [18]. Indeed, hybrid EMT phenotypes have been demonstrated to be both anoikis resistant and more resistant to therapy than cells that undergo complete EMT [42]. Moreover, the non-genetic heterogeneity observed in tumors displaying hybrid EMT can lead to differences in drug sensitivity and, over time, a subset of extremely drug-resistant cells [18]. One idea for treatment is to identify epigenetic regulators that can induce fully epithelial or mesenchymal phenotypes to elicit a more effective treatment response. This is complicated, however, by the fact that induction of EMT is context dependent, leading to the possibility of isogenic cells exhibiting different levels of EMT responsiveness to the same dose and duration of EMT-inducing stimuli [43]. Thus, before effective therapies can be developed, it will be important to understand the dynamics between the more epithelial-like and mesenchymal-like subpopulations. Intriguingly, gain and loss of E-cadherin can mediate population dynamics within a tumor to influence cell growth and behavior [44]. Thus, it is tempting to speculate that differential E-cadherin expression observed within the cSCC tumors may drive intercellular communication mechanisms that govern cell state acquisition.

The miR-200s/ZEB1 negative feedback loop is known to play an important role in the maintenance of an epithelial phenotype [45]. Herein, we demonstrate that the expression of miR-200b and miR-200c in cSCC tissues corresponds with heightened expression of ZEB1, especially in those cells that have lost their epithelial characteristics, which would not have been identified using cell lines or homogenized tissues. There is also significant in vitro and in vivo evidence indicating that EMT dynamics are influenced by cellular memory, and that this hysteretic control of EMT is governed by the miR-200s/ZEB1 negative feedback loop that was also observed in our cSCC tumor samples [46]. Hysteretic EMT enhances multi-stable EMT dynamics by increasing cell plasticity, resulting in a hybrid EMT phenotype that promotes metastatic efficiency. However, Ishay-Ronen et al. recently demonstrated that lung metastasis could be significantly reduced by converting breast cancer cells in a hybrid EMT state into post-mitotic adipocytes [47]. These findings suggest that it may also be possible to exploit cancer cell plasticity during hybrid EMT to improve treatment outcomes. Additionally, cell-fate determination between phenotypes is regulated by the miR-34/SNAI1 and miR-200s/ZEB1 negative feedback loops [48,49]. As these miR-34/SNAI1 and miR-200s/ZEB1 negative feedback loops have been shown to function as interconnected bistable switches that contribute to multi-stable EMT dynamics [48], our findings indicate that miRNA therapy may be a viable treatment approach for cSCC tumors exhibiting a hybrid EMT phenotype.

## 4. Materials and Methods

### 4.1. Patient-Derived Cutaneous Tissues

Tissue samples from patients presenting with cutaneous squamous cell carcinoma (cSCC) were acquired with informed consent via Mohs micrographic surgery at Affiliated Dermatology BioRepository (ALBR^®^) (Scottsdale, AZ, USA) as described by Belden et al. [50]. The single criterion for the collection of tissues is a biopsy-proven diagnosis of cSCC. The Institutional Review Board at Midwestern University approved the validation work using ALBR^®^ samples (MWU IRB Protocol AZ#807); samples from patients with a known blood-borne communicable disease were excluded from this study. Appendix A shows the demographic data of the patient-derived tissue samples used for all RNA and protein analyses. Immunofluorescence and immunohistochemistry staining were performed on commercially available cSCC tissue sections. Clinical diagnoses for the ALBR^®^-obtained cancerous tissue samples were categorized by dermatopathologists at Affiliated Dermatology as either non-invasive or invasive. Routine hematoxylin and eosin-stained sections were prepared at Affiliated Laboratories (Scottsdale, AZ, USA) and were interpreted by board-certified dermatopathologists at the same facility. Tumors were classified as invasive cSCC when there was clear dermal invasion of malignant keratinocytes. After the Mohs surgery, adjacent normal tissue (ANT) was removed from the wound margins during suturing. cSCC tissue and ANT samples intended for RNA extraction were immediately placed in RNAlater^®^ (Ambion Inc., Austin, TX, USA) and stored at −80 °C. Simultaneously, tissue samples intended for protein extraction were immediately stored at −80 °C.

### 4.2. Total Protein Isolation and Expression

Total protein was isolated from patient-derived cSCC and ANT by adding 500 µL RIPA buffer (50 nM Tris pH 8.0, 150 nM NaCl, 0.5% sodium deoxycholate, 1.0% NP-40, 0.1% SDS with protease inhibitors) per 100 mg tissue and homogenizing (BeadBug, Benchmark Scientific, Seville, Spain) at 20 s intervals for a total of 5 min at 4 °C. Samples were centrifuged at 21,000× *g* for 20 min, supernatants collected, and re-spun using the same conditions. Protein concentrations were determined using a BCA Protein Assay (Pierce) following the manufacturer’s protocol. A Proteome Profiler Human XL Oncology Array (R&D Systems, Minneapolis, MN, USA) to assess expression levels of 84 human cancer-related proteins was performed on 6 cSCC and 6 ANT samples, in duplicate, using 400 μg of protein following the manufacturer’s protocol. Overall expression of each protein was measured and normalized to vinculin (Appendix A).

### 4.3. Fluorescence Resonance Energy Transfer (FRET) MMP Assay

Two hundred and fifty micrograms of total protein isolated from patient-derived cSCC tissue and ANT samples was used to perform a total MMP FRET assay (AnaSpec, Inc., Fremont, CA, USA) following the manufacturer’s recommendation. Briefly, 250 µg of each protein sample was incubated with 1 mM aminophenylmercuric acetate (APMA) at 37 °C for 60 min to activate MMPs, after which 50 μL of a 1:100 dilution of total MMP substrate solution was added and transferred, in triplicate, to a 96-well plate. The fluorescence signal was measured in a kinetic assay at 5 min intervals for 60 min, then at 2.5 and 4 h, using a BioTek Cytation3 plate reader (BioTek, Winooski, VT, USA). Positive (total MMP substrates) and negative (buffer) controls were also included. Relative MMP activity over time, represented by the maximum velocity, was calculated, and graphed using Microsoft Excel.

### 4.4. Western Blotting

Twenty to forty micrograms of each protein sample and control protein (either P-cadherin or HeLa cell lysate) were resolved on a 4–20% Mini-PROTEAN^®^ TGX™ Precast Protein gel (Bio-Rad, Hercules, CA, USA) based on their molecular weights, transferred to a PVDF membrane, and blocked using 5% non-fat dry milk (NFDM, 1X TBS, 0.1% Tween 20) for one hour. Primary antibodies used: 1:1000 rabbit anti-phospho (S838 + S840) E-cadherin (Abcam, Cambridge, UK), 1:500 rabbit anti-E-cadherin (Abcam, Cambridge, UK), 1:250 mouse anti-vimentin (Santa Cruz Biotechnology [SCBT]), 1:500 rabbit anti-TGF-β (anti-TGFβ1/2/3; SCBT), and 1:1000 rabbit anti-GAPDH (Cell Signaling Technology) in non-fat dry milk. Primary antibodies were omitted as a negative control. A 1:10,000 dilution of either AlexaFluor^®^ 790 (Abcam, Cambridge, UK) or an HRP-conjugated secondary antibody was used. All blots were performed in triplicate and relative expression was measured using either an Odyssey^®^ CLx (LI-COR Biotechnology, Lincoln, NE, USA) or ChemiDoc XRS + (Bio-Rad, Hercules, CA, USA) imaging system. Band intensities of phosphorylated E-cadherin, E-cadherin, vimentin, and TGF-β were normalized to GAPDH and analyzed using ImageJ (NIH).

### 4.5. RNA Isolation and Quantitative Real-Time PCR

Frozen tumor tissue samples (ranging from 2.0–16.6 mg) and adjacent normal tissue samples (ranging from 6.0–33.0 mg) were trimmed to remove the subcutaneous layer and homogenized with a motorized handheld rotor. Extraction of total RNA was performed using TRIzol^®^ (Ambion Inc., Austin, TX, USA) following the manufacturer’s protocol with the addition of 100% isopropanol overnight at −80 °C to maximize RNA yield. The quality, concentration, and purity of each RNA sample were determined using a NanoDrop 1000 spectrophotometer (Thermo Scientific, Waltham, MA, USA) and 2% agarose gel electrophoresis. RNA samples collected from tissues that yielded intact RNA with a 260/280 ratio greater than 1.80 and a 260/230 ratio greater than 1.60 only were used in all downstream applications. DNase-treated RNA was reverse-transcribed into cDNA using SuperScript II (Life Technologies, Carlsbad, CA, USA). Real-time PCR was performed using either an ABI StepOnePlus™ thermocycler or Bio-Rad CFX thermocycler using Power SYBR^®^ Green (Applied Biosystems, Waltham, MA, USA). Primer sequences used were as follows: E-cadherin—Forward: 5′- CCC GGG ACA ACG TTT ATT AC-3′, Reverse: 5′- GCT GGC TCA AGT CAA AGT CC-3′ [51]; TGF-β—Forward: 5′-TCC TGG CGA TAC CTC AGC AA-3′, Reverse: 5′- CTC AAT TTC CCC TCC ACG GC-3′ [52]; TGF-βR2—Forward: 5′- AAT GTG AAG GTG TGG AGA C-3′, Reverse: 5′- GGT AGG CAG TGG AAA GAG-3′ [53] Cyclophilin—Forward: 5′- TGC CAT CGC CAA GGA GTA-3′, Reverse: 5′- TGC ACA GAC GGT CAC TCA AA-3′. A total of 5 cSCC and 5 ANT patient-derived RNA/cDNA samples were used for all genes tested in triplicate. Relative expression of each target gene was determined using the 2^−ΔΔCT^ method using cyclophilin as the reference control.

### 4.6. miRNA Analysis

A total of eight miRNA primers (miR-21, -31, -34a, -181, -200a, -200b, -200c, and -205) purchased as miScript Primer Assays (proprietary to Qiagen) were rehydrated per the manufacturer’s recommendations using 550.0 mL 1X Tris-EDTA. RNU6, a small non-coding RNA (snRNA), was used as our reference RNA and was also purchased from Qiagen. A total of 6 cSCC and 6 ANT patient-derived samples were used. For miRNA analysis, 250 ng of total RNA from each tissue sample was reverse transcribed into cDNA using the miScript II Reverse Transcription kit (Qiagen, Hilden, Germany) following the manufacturer’s recommended protocol. For each cDNA sample, a qPCR reaction was performed in duplicate for each miRNA and reference RNA (RNU6). All reactions were performed in 96-well plates and amplification was detected using miScript SYBR^®^ Green (Qiagen, Hilden, Germany) and a Bio-Rad CFX thermocycler (Applied Biosystems, Waltham, MA, USA). After each qPCR reaction, the relative expression of each miRNA as compared to RNU6 was calculated using 2^−^^ΔΔCT^ where ΔΔC_T_ = ((C_T_ − RNU6) − maxΔC_T_). Fold-change in miRNA expression was graphed and analyzed in Excel using a Welch’s *t*-test.

### 4.7. Indirect Immunofluorescence Assays

Slides of formalin-fixed, paraffin-embedded cSCC tissue and ANT sections were purchased (BioChain Institute Inc, Newark, CA, USA) and a standard immunofluorescence protocol was performed. Our patient-derived tissue samples, obtained by Mohs surgery, were unsuitable for sectioning. Briefly, sections were baked at 60 °C for 60 min, then de-paraffinized by placing in xylene followed by reducing concentrations of ethanol (100 to 70%). Heat-induced epitope retrieval (HIER) was performed using either citrate buffer (pH 6.0) for E-cadherin, phosphorylated E-cadherin, and β-catenin or basic buffer (pH 9.0) for ZEB1 and SNAI1 in conjunction with pan-cytokeratin following the manufacturer’s protocol. De-paraffined sections were permeabilized using 0.25% Trypsin and blocked with 1X-TBS containing 0.1% Tween-20 and 10% normal goat serum for 1 hr at room temperature. After blocking, cells were incubated overnight with either 1:50 dilution of mouse anti-E-cadherin (SCBT, Dallas, TX, USA) + 1:200 dilution of rabbit anti-β-catenin (Abcam, Cambridge, UK), 1:100 dilution of rabbit anti-phospho-E-cadherin (Abcam, Cambridge, UK), 1:100 dilution of rabbit anti-SNAI1 (Novus Biological, Arapahoe, CO, USA), or 1:100 dilution of rabbit anti-ZEB1 (Novus Biological, Arapahoe, CO, USA) + 1:100 dilution of mouse monoclonal antibody to pan-cytokeratin C-11 (SCBT, Dallas, TX, USA). Primary antibodies were omitted as a negative control. The tissue sections were washed with 1X-TBST, then incubated with a 1:250 dilution of either a goat anti-mouse-488 and/or 594 or a goat anti-rabbit-488 and/or 594 secondary antibodies (Invitrogen, Waltham, MA, USA), respectively, for 1 hr. Slides were overlaid with a DAPI-containing mounting medium. Images were obtained at 100× and 630× magnification using a confocal microscope (Leica). Protein expression was measured using ImageJ (NIH); the region of interest was defined using the selection tools and the selected area, integrated density, and mean gray value were measured. Total cell fluorescent intensity was calculated from the corrected mean intensity multiplied by the area. For all images, analyzed exposures were kept constant and the data analyzed using Microsoft Excel.

### 4.8. Immunohistochemistry Assays

Slides of formalin-fixed, paraffin-embedded cSCC tissue and ANT sections were purchased (US Biomax, Derwood, MD, USA) and a standard immunohistochemistry protocol was performed. Briefly, sections were baked at 60 °C for 60 min, then de-paraffinized by placing in xylene followed by reducing concentrations of ethanol (100% to 70%). Heat-induced epitope retrieval (HIER) was performed using citrate buffer (pH 6.0) following the manufacturer’s protocol. De-paraffined sections were permeabilized using 0.25% Trypsin without EDTA for 10 min and blocked with 1X-TBS containing 0.1% Tween-20 and 10% goat serum for 1 h at room temperature. After blocking, cells were incubated overnight with either a 1:50 dilution of mouse anti-vimentin (SCBT, Dallas, TX, USA) or a 1:200 dilution of rabbit anti-pan-cytokeratin (Abcam, Cambridge, UK). The tissue sections were washed with 1X-TBST, then incubated with a 1:250 dilution of either a goat anti-rabbit IgG-AP (SCBT, Dallas, TX, USA) or a goat anti-mouse IgG-AP (SCBT, Dallas, TX, USA) secondary antibody for 1 hr. Slides were developed using permanent red substrate, counterstained with Mayer’s hematoxylin, and overlaid with paramount G mounting medium. Isotype controls were performed to ensure absence of non-specific antibody binding. Images were obtained at 40× and 100× magnifications using an inverted brightfield microscope (Olympus CKX41). Protein expression was measured using ImageJ/Fiji (NIH). For each image, color deconvolution for red staining was performed, and maximum intensity and mean gray values were measured. Optical density was calculated by taking the log of maximum intensity divided by mean intensity and analyzed using Microsoft Excel.

### 4.9. Statistical Analyses

Statistical differences in RNA expression between samples were determined used Welch’s *t*-test (Microsoft Excel). Outliers within the dataset were calculated using the ROUT method with Q = 1% (GraphPad Prism 7.0); one outlier was identified and removed from the dataset before analysis. Statistical differences in protein expression on Western blots, immunofluorescence assays, and immunohistochemistry staining between ANT and cSCC samples were determined using a paired *t*-test (Microsoft Excel). Additionally, the amount of staining between poorly differentiated and well-differentiated cells in the IHC images was analyzed semi-quantitatively using a 0–3+ scale, with 0 = no staining, 1+ = <10% positive staining, 2+ = 10–50% positive staining, and 3+ = >50% positive staining.

## 5. Conclusions

Within a heterogeneous mass of cancer cells, there may be variation in the extent or stage of EMT; some cells may retain nearly all the traits associated with epithelial cells, some cells may express a mixture of epithelial and mesenchymal proteins, and some cells may appear to become fully mesenchymal, with little to no expression of their original epithelial characteristics. Our data collectively support that during development of cSCC, there is differential expression of the EMT program within the cancerous lesions. The cells at the leading edges of the tumor exhibit more advanced EMT, possibly due to their proximity to the extracellular matrix, where TGF-β levels and MMP activity are higher. Cells within the tumor mass may not be stimulated to the same extent, and therefore are either lagging behind in their EMT program, or there is some unknown collective cell signaling occurring between the poorly differentiated and well-differentiated cells. This hybrid EMT program may be promoting survival and invasiveness of the tumors as evidenced by the higher expression of both epithelial and mesenchymal markers in invasive cSCC tissues compared to non-invasive cSCC tissues. A better understanding of how this hybrid EMT process is regulated or controlled may influence the therapeutic strategies in patients with more invasive disease.

## Figures and Tables

**Figure 1 ijms-23-09183-f001:**
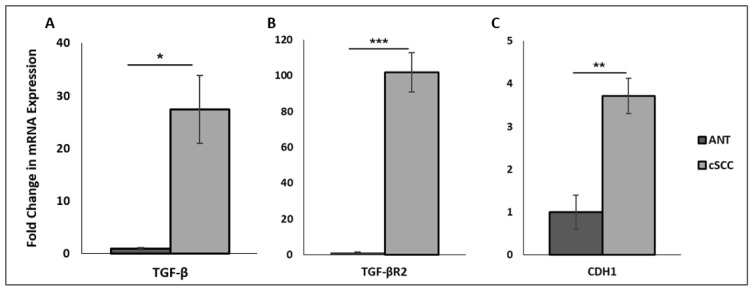
Fold change in TGF-β (Panel **A**), TGF-βR2 (Panel **B**), and E-cadherin (CDH1; Panel **C**) mRNA expression between ANT and cSCC tissues. All values are expressed as mean ± SEM. * denotes *p* < 0.05, ** denotes *p* < 0.01, and *** denotes *p* < 0.001.

**Figure 2 ijms-23-09183-f002:**
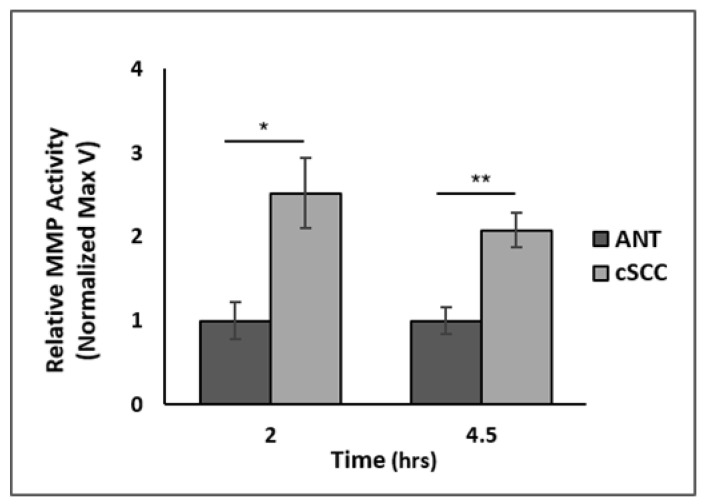
Fluorescence resonance energy transfer (FRET) assay of MMP activity in tissue samples. Cell lysates from homogenized ANT and cSCC tissues were used to assess relative MMP activity, measured at two different time points after MMP activation. All values are expressed as mean ± SEM. * denotes *p* < 0.05 and ** denotes *p* < 0.01.

**Figure 3 ijms-23-09183-f003:**
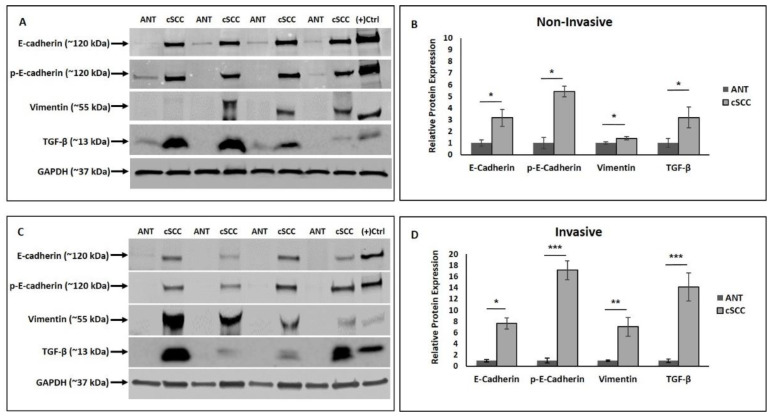
Western blot analysis of the epithelial markers E-cadherin and phosphorylated E-cadherin, the mesenchymal marker vimentin, and TGF-β protein expression in non-invasive (Panels **A**,**B**) and invasive (Panels **C**,**D**) cSCC tissues compared to ANT. Positive control lane: P-cadherin for the cadherins and HeLa cell lysate for vimentin and TGF-β. All values are expressed as mean ± SEM. * denotes *p* < 0.05, ** denotes *p* < 0.01, and *** denotes *p* < 0.001.

**Figure 4 ijms-23-09183-f004:**
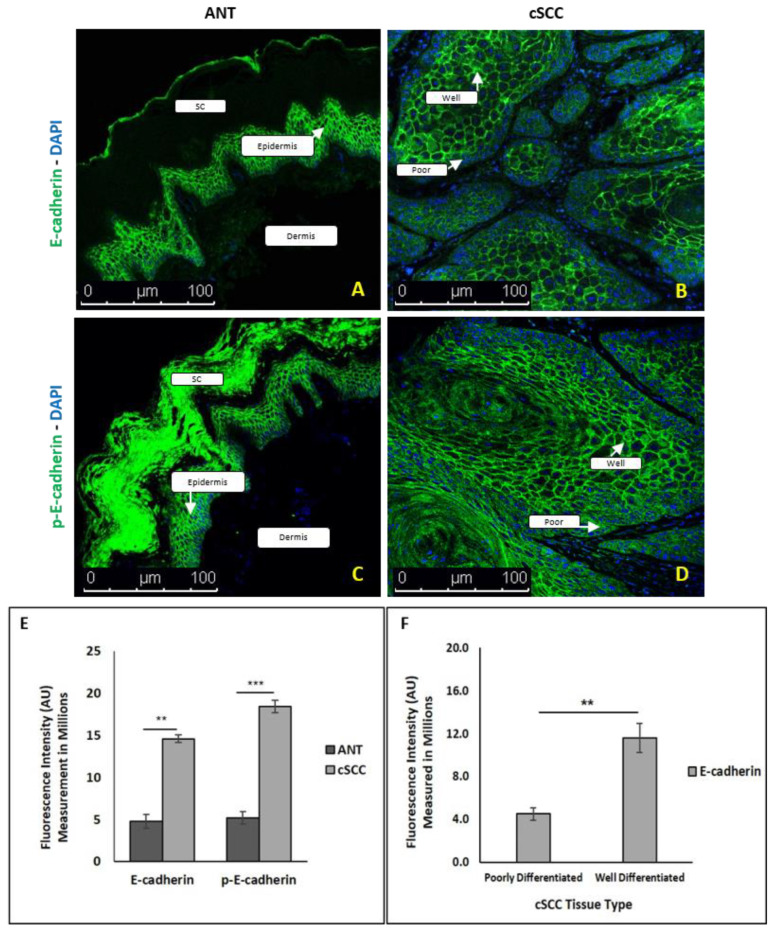
Immunofluorescence staining of ANT and cSCC tissues for E-cadherin (Panels **A**,**B**) and phosphorylated E-cadherin (Panels **C**,**D**); images were taken at 100× magnification. The epidermis and dermis layers of ANT are labeled, with the stratum corneum (sc) layer at the top. Quantitative analysis of protein expression between ANT and cSCC (Panel **E**) and E-cadherin expression between well-differentiated tumor cells (well) in the center of the tumor mass vs. poorly differentiated tumor cells (poor) in the leading edge of the tumor (Panel **F**) was performed. All values are expressed as mean ± SEM. ** denotes *p* < 0.005 and *** denotes *p* < 0.0001.

**Figure 5 ijms-23-09183-f005:**
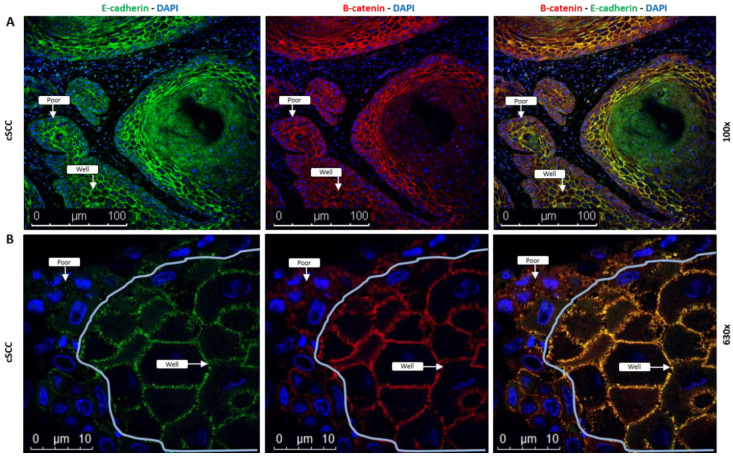
Immunofluorescence staining of cSCC tissues for E-cadherin (green) and β-catenin (red). The well-differentiated cancer cells (well) show that E-cadherin and β-catenin appear to be coexpressed (yellow) with loss of this coexpression in the poorly differentiated cells (poor) along the leading edges of the tumors. Images in Panel **A** (top) were taken at 100× magnification and show the well-differentiated cells (well) central to the poorly differentiated cells within each tumor mass. Images in Panel **B** (bottom) were taken at 630× magnification; the blue solid line delineates the poorly differentiated cells from the well-differentiated cells within a tumor mass, with the cells to the right being toward the center of the tumor mass.

**Figure 6 ijms-23-09183-f006:**
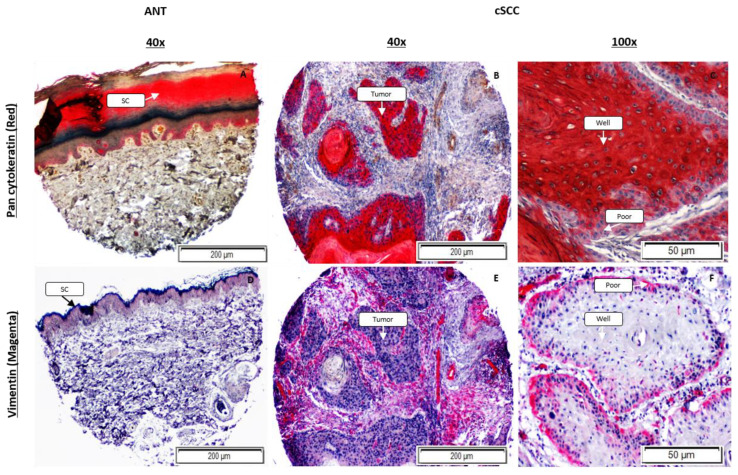
Immunohistochemistry of ANT and cSCC tissues for pan-cytokeratin (red; Panels **A**–**C**) and vimentin (magenta; Panels **D**–**F**). At 40× magnification, cSCC images delineate cSCC (tumor) tissue from the dermal tissue and ANT images are oriented with the stratum corneum (sc) layer at the top. Images at 100× magnification show that poorly differentiated cells (poor) along the edges of the tumor have a higher (2–3+) expression of the mesenchymal marker vimentin (Panel **F**) compared to well-differentiated cells (well; 0–1+) in the center of the tumor mass. Poorly differentiated cells also show a visible reduction in the epithelial marker pan-cytokeratin (Panel **C**). Tissue sections were stained with H&E for validation of epithelial tissue (images not shown).

**Figure 7 ijms-23-09183-f007:**
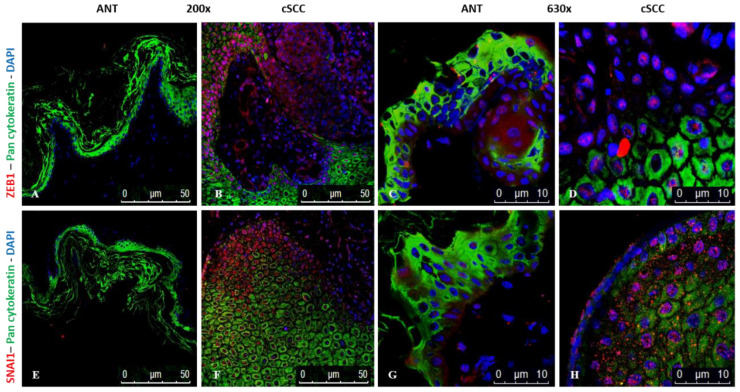
Immunofluorescence showing expression of ZEB1 (red; Panels **A**–**D**) and SNAI1 (red; Panels **E**–**H**) in ANT and cSCC tissues. In all cSCC images (Panels **B**,**D**,**F**,**H**), the poorly differentiated cells at the edges of a tumor mass are oriented toward the top and left side of the images with the well-differentiated cells underneath and to the right. Cytokeratin expression (green) was used to delineate well-differentiated epithelial cells in all panels. Blue = DAPI.

**Figure 8 ijms-23-09183-f008:**
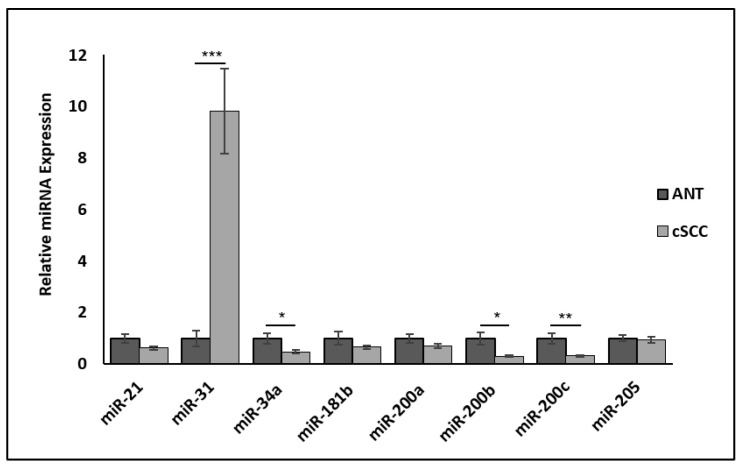
Differential expression of four oncogenic and EMT-related miRNAs between ANT and cSCC tissues. * denotes *p* < 0.05, ** denotes *p* < 0.01, and *** denotes *p* < 0.001.

## Data Availability

Not applicable.

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
