# Peer review of "A Hybrid Epithelial to Mesenchymal Transition in Ex Vivo Cutaneous Squamous Cell Carcinoma Tissues"

_ijms, 2022, doi:10.3390/ijms23169183_

Round 1
Reviewer 1 Report
The paper by Pulford et al. described the impact of EMT in Cutaneous Squampus Cell Carcinoma (cSCC) showing the presence of a hybrid EMT program in cancer cells. The novelty of this paper is limited because EMT as master regulator of cSCC aggressivness was already described in literature. The authors showed that there is a phenotype plasticity within the tumor tissue, an emerging concept in different cancer models. To publish this paper I suggest to the authors to improve the results:
1. Result 2.1: add evaluation of other markers of EMT such as Twist, Zeb1, NCAD, Snail (mRNA and protein levels). It would be better to evaluate also the activation of TGF-b signaling pathway involved in EMT (for example SMAD activation).
2. Result 2.2: it would be better to show as a heatmap the differentiated oncogenic proteins in cSCC vs ANT
3. Figure 3: improve the resolution of WB
4. Result 2.7: Add WB analysis of ZEB1 and SNAIL1 in cells
5. Result 2.8: the connection between EMT proteins and their regulation by miRs is very interesting but here the authors showed only the expression of some miRs comparing cSCC and ANT (only a descriptive data). Which is the impact of these miRs on EMT? The authors should investigate, selecting a tumor-suppressor miR and an oncomiR, the impact of manipulation of these miRs (overexpressing tumor-suppressor miR and silencing oncomiR) on EMT protein expression and EMT process (TGF-b signaling pathway activation, invasiveness of cells).
Author Response
We thank the reviewer for the valuable comments and have addressed them as best as we could. The following are our responses to each comment:
- Result 2.1: add evaluation of other markers of EMT such as Twist, Zeb1, NCAD, Snail (mRNA and protein levels). It would be better to evaluate also the activation of TGF-b signaling pathway involved in EMT (for example SMAD activation).
- We appreciate the suggestion of additional experiments by the reviewer and acknowledge that these results would provide further evidence of EMT occurring in the cSCC tissues. However, as our primary hypothesis was that there is phenotypic plasticity within the tumor tissue, as this reviewer noted, we feel these additional experiments do not support our primary hypothesis. In addition, due to our limitation in acquiring additional clinical samples (which took several months), additional experiments of this nature cannot be performed with the 10-day deadline we were given to revise and resubmit.
- Result 2.2: it would be better to show as a heatmap the differentiated oncogenic proteins in cSCC vs ANT
- We greatly appreciated this suggestion and investigated creating a heat map. In looking at expression of all 84 genes, essentially all of them are expressed either the same as or higher than in ANT, which would be expected in an oncogenic profiler array of cancerous versus normal tissue. In creating this heat map, what we got was a map that just showed some proteins were expressed higher (i.e., a darker red color) vs other proteins (i.e., a lighter red or white) in comparison to ANT (i.e., white), which would be the baseline used for comparison. We feel that the current bar graph showing the 13 genes that are overexpressed in cSCC vs ANT is clearer than the heat map we generated. However, if this reviewer or the editors wish to see the heat map we attempted to generate and determine if it should also be included as supplementary data, we would be happy to submit it for your review.
- Figure 3: improve the resolution of WB
- We apologize that this WB was not as high of a resolution as possible. We have improved the resolution and provided this updated image in the revised manuscript after line 114 (top of page 4). The images attached in separate files may show this increased resolution better than in the image embedded in the body of the manuscript, and the .tif image we will provide is at a higher resolution.
- Result 2.7: Add WB analysis of ZEB1 and SNAIL1 in cells
- We are unable to perform additional experimentation at this time due to our limitation in acquiring more clinical samples and completing experimentation within the 10-day time frame we were given to revise and resubmit. As these proteins likely are not expressed very abundantly within the tissues, we opted to show expression of these two transcription factors using immunofluorescence and immunohistochemistry (IHC) instead of using WB analysis. We have quantified the IF data in Figure 7 and included the results within the text (lines 184-188) and also included our quantified IHC data as Supplementary Figure 3 (see lines 496-497) and in the text (lines 195-196). We hope the reviewer will agree there is a significant difference in expression of these transcription factors between ANT and cSCC cells.
- Result 2.8: the connection between EMT proteins and their regulation by miRs is very interesting but here the authors showed only the expression of some miRs comparing cSCC and ANT (only a descriptive data). Which is the impact of these miRs on EMT? The authors should investigate, selecting a tumor-suppressor miR and an oncomiR, the impact of manipulation of these miRs (overexpressing tumor-suppressor miR and silencing oncomiR) on EMT protein expression and EMT process (TGF-b signaling pathway activation, invasiveness of cells).
- We agree that these suggested experiments that would strengthen the contribution of these miRs and thank the reviewer for this suggestion. However, we are unable to conduct these types of manipulative experiments using patient-derived tissues or commercially available, fixed tissue sections. However, these experiments can be performed using an established cSCC cell line(s) and we are considering doing so soon. However, we feel these experiments are outside the scope of this current manuscript using patient-derived tissues as well as would take significantly longer than the 10-day deadline we were given to revise and resubmit.
Reviewer 2 Report
The paper of Pulford CS et al talks about the differential expression of EMT programs in cutaneous squamous cell carcinomas, which they suggest to be correlated to an aggressive disease course. The concept of the study is good, however, technically there are several issues to be addressed.
Minor comments:
· Lane 31: The abbreviation cSCC is introduced for squamous cell carcinoma. However, “cutaneous” is missing.
Major comments:
· Figure 3: The positive control is not described, and the negative control is missing.
· Figure 4: Pictures of immunofluorescence stainings of ANT shoult be presented with the stratum corneum up. Sections should be labelled well. Also, it should be indicated, fluorescence intensity of which region was measured and used for calculations. Distinct regions look overexposed.
· Section 2.5 should include a statement with a rational for the respective stainings and a context-embedded description of results.
· Figure 5: Pictures should be labelled to guide the reader to relevant regions (e.g. leading edges). It is not clear to me what is shown in panel B, which should be a 6.4x magnification of panel A.
· Figure 6: Normal skin sections are upside down. Again, relevant regions should be indicated.
· Figure 7: Same as above. Green fluorescence seems to be in the saturation.
· Tissues: It is not clear how tissues were classified. Authors say that the only criteria was diagnosis of cSCC. No sections could be done and invasiveness was done by dermatopathologists. How was this done? Were sections available from the provider? Overall, H&E stainings should be shown.
· Controls, at least negative (i.e. second step), should be shown for immunofluorescence stainings.
· Authors state that outliers were calculated and excluded. How many outliers were identified?
· For Western blot and Real-time PCR: indicate how many biological and technical replicates were performed.
Author Response
We thank the reviewer for the valuable comments and have addressed them as best as we could. The following are our responses to each comment:
Minor comments:
- Lane 31: The abbreviation cSCC is introduced for squamous cell carcinoma. However, “cutaneous” is missing.
- We apologize for omitting the word “cutaneous” on line 31 and have added it into the manuscript.
Major comments:
- Figure 3: The positive control is not described, and the negative control is missing.
- We appreciate this comment and have edited the figure legend (lines 117-118) and the text in the methods (lines 373-374) to state the positive controls (P-cadherin and HeLa cell lysates) we used for the Western blots. We used secondary antibody only (omitting primary antibody) as our negative control to ensure that we did not have any non-specific binding of our secondary antibody on the blots and included this statement on lines 380-381.
- Figure 4: Pictures of immunofluorescence stainings of ANT shoult be presented with the stratum corneum up. Sections should be labelled well. Also, it should be indicated, fluorescence intensity of which region was measured and used for calculations. Distinct regions look overexposed.
- We appreciate this comment and we have made the adjustments and added labels to the figures suggested by this reviewer. We also included a description of how measurements were obtained in the methods section on lines 444-448. By keeping exposures constant and at a level to show measurable levels of expression, this resulted in the keratinized layer in ANT Panel B being overexposed. However, this layer was not used to collect any measurements.
- Section 2.5 should include a statement with a rational for the respective stainings and a context-embedded description of results.
- We appreciate this comment and apologize for the omission for the rationale and context-dependent description of these results. We have edited the manuscript (lines 140-146) to provide this information and clarify our interpretation of the results.
- Figure 5: Pictures should be labelled to guide the reader to relevant regions (e.g. leading edges). It is not clear to me what is shown in panel B, which should be a 6.4x magnification of panel A.
- We appreciate this comment and have added labels to the figures suggested by this reviewer and revised the figure legend (lines 152-158) to guide the reader to the relevant regions more clearly. This reviewer is correct that panel B is an increased magnification of panel A as indicated on lines 147-148 and 154-156.
- Figure 6: Normal skin sections are upside down. Again, relevant regions should be indicated.
- We appreciate this comment and we have made the adjustments and added labels to Figure 6 as suggested by this reviewer. We have also updated the legend for Figure 6 (lines 173-180).
- Figure 7: Same as above. Green fluorescence seems to be in the saturation.
- We appreciate this comment and we have made the adjustments and added labels to Figure 7 as suggested by this reviewer. We used the green as a counterstain to show which cells were epithelial-like, which separated our leading-edge (poorly differentiated) cells from the well differentiated cells in the center of the tumor mass. We kept exposures constant; this exposure was a trade-off to allow visual detection and quantification (quantification added per another reviewer’s suggestion, see lines 184-188) of ZEB1 and SNAI1 expression within the cells. As we did not quantitate the green fluorescence, we feel this trade-off in exposure is warranted. However, to address another reviewer’s comment, we are also providing our IHC data as Supplementary Figure 3 (see lines 496-497) and included results on lines 195-196 to further clarify the expression of these transcription factors within the cells.
- Tissues: It is not clear how tissues were classified. Authors say that the only criteria was diagnosis of cSCC. No sections could be done and invasiveness was done by dermatopathologists. How was this done? Were sections available from the provider? Overall, H&E stainings should be shown.
- We appreciate this reviewer’s comment and have provided clarification on the criteria used in classification of tumors and verification of H&E staining performed at Affiliated Dermatology in the methods section (lines 341-344).
- Controls, at least negative (i.e. second step), should be shown for immunofluorescence stainings.
- We used secondary antibody only (omitting primary antibody) as our negative control to ensure that we did not have any non-specific binding of our secondary antibody in these images. We have edited the manuscript on line 439 to provide this information.
- Authors state that outliers were calculated and excluded. How many outliers were identified?
- We appreciate this reviewer catching this omission on our part and we have subsequently clarified on lines 469-470 that we excluded only one outlier from our data set.
- For Western blot and Real-time PCR: indicate how many biological and technical replicates were performed.
- We provided the number of biological and technical replicates for our mRNA qPCR data and WB data within the text of the results section (lines 76-80 and 106-113, respectively), but we realize this was omitted for our miRNA qPCR data in both the methods and results sections for Figure 8. We have added the number of biological replicates used for miRNA qPCR within the results (lines 209, 211) and in the methods (lines 413-414). For each of our WB and qPCR experiments, we performed technical replicates in either duplicate or triplicate, and these are included on lines 382, 405-406 and 417-418 in the corresponding methods sections.
Reviewer 3 Report
This manuscript reports the analysis of EMT markers in cutaneous squamous cell carcinomas (SCC) and their association with tumor cell differentiation and invasiveness. Based on these analyses the authors conclude that a hybrid EMT process operates during SCC progression that may contribute to tumor invasion. The dynamics of the EMT process during tumor development is not well understood and new insights are needed. However, most of the observations described in this manuscript have been previously reported. In addition, the cohort studied here contains invasive and non-invasive tumors, but it is unclear how the hybrid EMT phenotypes affect each group. Lastly, some of the claims are not well supported by the data presented, particularly the criteria used to identify poorly differentiated areas of the tumors in an unbiased manner. Specific comments:
1. Describe the criteria used to classify tumors as invasive or non-invasive. Histological images of each group would be helpful to illustrate the criteria.
2. Fig 3. Side by side comparison of non-invasive vs invasive tumors would be needed to support the claim that changes in the expression of these proteins is different in invasive and non-invasive tumors. The expression of most of these proteins is extremely low in ANT and therefore fold changes are not very reliable to support this claim. Also, it is intriguing that the expression of E-cadherin is very low to non-detectable by western blot in many ANT tissues. However, E-cadherin is shown to be highly expressed in the ANT epidermis (Fig 4). Epithelial cellularity in adjacent skin vs SCCs should be taken into consideration as it is possible that the increase in E-cadherin observed in SCCs could be due to an increase in epithelial cells in tumors, rather than to an increase in expression. This should be discussed.
3. Figs 4-7. Describe the criteria used to identify well and poorly differentiated areas of the tumor. It seems that in many of the images shown in these figures the only criteria used is the expression of the markers analyzed. Also, the data described in these figures need to be quantified.
4. Fig 5. The authors claim a decrease in E-cadherin and b-catenin co-localization in poorly differentiated areas of the tumors. First, it unclear how well and poorly differentiated areas are defined. Second, there seems to be a decrease in co-expression of these protein in certain areas of the tumor, rather than reduced co-localization. To support these claims the authors should quantify co-localization and clearly indicate what areas of the tumors are considered well and poorly differentiated.
Author Response
We thank the reviewer for the valuable comments and have addressed them as best as we could. The following are our responses to each comment:
- Describe the criteria used to classify tumors as invasive or non-invasive. Histological images of each group would be helpful to illustrate the criteria.
- We appreciate this reviewer’s comment and have provided clarification on the criteria used in classification of tumors and verification of H&E staining performed at Affiliated Dermatology in the methods section (lines 341-344).
- Fig 3. Side by side comparison of non-invasive vs invasive tumors would be needed to support the claim that changes in the expression of these proteins is different in invasive and non-invasive tumors. The expression of most of these proteins is extremely low in ANT and therefore fold changes are not very reliable to support this claim. Also, it is intriguing that the expression of E-cadherin is very low to non-detectable by western blot in many ANT tissues. However, E-cadherin is shown to be highly expressed in the ANT epidermis (Fig 4). Epithelial cellularity in adjacent skin vs SCCs should be taken into consideration as it is possible that the increase in E-cadherin observed in SCCs could be due to an increase in epithelial cells in tumors, rather than to an increase in expression. This should be discussed.
- We value these comments by this reviewer and realize that we did state that expression of these proteins is higher in invasive vs non-invasive tissues. The reviewer is correct that we would needed to have compared E-cadherin expression of invasive vs non-invasive on the same blots to support our initial claims, so we have rephrased the text accordingly (deleted lines 109-110 and edited lines 257-258 in the discussion). However, E-cadherin expression is not “very low” in the ANT samples. Rather, it is so high in the cSCC samples that exposure times were kept very low to keep from saturating the pixels, which would have prevented us from accurately quantifying the data. This is evidenced, as the reviewer states, in Figure 4, where E-cadherin expression is quite abundant in the ANT samples, but these images were optimized for detection to demonstrated localization in these tissues. Had we tried to be quantitative with the IF images and adjusted for exposure time, E-cadherin levels would have looked much lower than in the cSCC samples.
- Figs 4-7. Describe the criteria used to identify well and poorly differentiated areas of the tumor. It seems that in many of the images shown in these figures the only criteria used is the expression of the markers analyzed. Also, the data described in these figures need to be quantified.
- We appreciate this comment and have edited the figures to delineate the well differentiated and poorly differentiated areas of the images. We did quantify data in Figure 4 and the results are in Figure 4 and in lines 124-125; we added in the quantification of Figure 6 and edited the results section (lines 170-171), figure legend (lines 173-180), and the methods section (lines 472-475). We quantified the data in Figure 7 and provided those results in the text (lines 184-188). We also are providing IHC staining for these transcription factors as Supplementary Figure 3 and added in the results (lines 195-196) that provides additional, quantifiable data for ZEB1 and SNAI1 expression to support the IF data in Figure 7. We discuss Figure 5 more specifically in the reviewer’s next comment, below.
- Fig 5. The authors claim a decrease in E-cadherin and b-catenin co-localization in poorly differentiated areas of the tumors. First, it unclear how well and poorly differentiated areas are defined. Second, there seems to be a decrease in co-expression of these protein in certain areas of the tumor, rather than reduced co-localization. To support these claims the authors should quantify co-localization and clearly indicate what areas of the tumors are considered well and poorly differentiated.
- We chose not to quantify the data in Figure 5 as we are not intending to show changes in protein expression but a visible loss of co-localization of E-cadherin and β-catenin in the poorly differentiated cells, providing support that the cells at the leading edge are losing some of their adherence properties. However, we do appreciate and acknowledge this reviewer’s comment that we are assuming co-localization due to the overlap in protein expression detected by the yellow fluorescent color and cannot distinguish between colocalization and coexpression. We have edited the figure legend (lines 151-158) and text in the results (lines 144-146) and discussion (lines 270 and 282) to clarify the points we are making regarding these data.
Round 2
Reviewer 1 Report
Due to a limited time for revision some additional experiments did not performed by the authors. I hope that in the future the authors will work on some points that I highlighted. However I think that the revised manuscript was improved and can be accepted for publication.
Author Response
We again thank this reviewer for their time in conducting a careful read through the manuscript and for approving our submitted changes. We do have plans to complete some of the additional experiments suggested by this reviewer and appreciate that this reviewer will find those valuable.
Reviewer 2 Report
I have no more comments. All points raised in the first review were addressed.
Author Response
We thank this reviewer for a careful read through our resubmission and are pleased that we addressed all comments to this reviewers satisfaction.